# High Platelet Reactivity after Transition from Cangrelor to Ticagrelor in Hypothermic Cardiac Arrest Survivors with ST-Segment Elevation Myocardial Infarction

**DOI:** 10.3390/jcm9020583

**Published:** 2020-02-21

**Authors:** Nina Buchtele, Harald Herkner, Christian Schörgenhofer, Anne Merrelaar, Roberta Laggner, Georg Gelbenegger, Alexander O. Spiel, Hans Domanovits, Irene Lang, Bernd Jilma, Michael Schwameis

**Affiliations:** 1Department of Clinical Pharmacology, Medical University of Vienna, 1090 Vienna, Austria; nina.buchtele@meduniwien.ac.at (N.B.); christian.schoergenhofer@meduniwien.ac.at (C.S.); georg.gelbenegger@meduniwien.ac.at (G.G.); bernd.jilma@meduniwien.ac.at (B.J.); 2Department of Medicine I, Medical University of Vienna, 1090 Vienna, Austria; 3Department of Emergency Medicine, Medical University of Vienna, 1090 Vienna, Austria; harald.herkner@meduniwien.ac.at (H.H.); anne.merrelaar@meduniwien.ac.at (A.M.); hans.domanovits@meduniwien.ac.at (H.D.); michael.schwameis@meduniwien.ac.at (M.S.); 4Department of Orthopedics and Trauma-Surgery, Medical University of Vienna, 1090 Vienna, Austria; roberta.laggner@meduniwien.ac.at; 5Department of Medicine II, Division of Cardiology, Medical University of Vienna, 1090 Vienna, Austria; irene.lang@meduniwien.ac.at

**Keywords:** transition, cangrelor, ticagrelor, P2Y12, cardiopulmonary resuscitation, hypothermia

## Abstract

Transition from cangrelor to oral P2Y12 inhibitors after PCI carries the risk of platelet function recovery and acute stent thrombosis. Whether the recommended transition regimen is appropriate for hypothermic cardiac arrest survivors is unknown. We assessed the rate of high platelet reactivity (HPR) after transition from cangrelor to ticagrelor in hypothermic cardiac arrest survivors. Adult survivors of out-of-hospital cardiac arrest with ST-segment elevation myocardial infarction (STEMI), who were treated for hypothermia (33 °C ± 1) and received intravenous cangrelor during PCI and subsequent oral loading with 180mg ticagrelor were enrolled in this prospective observational cohort study. Platelet function was assessed using whole blood aggregometry. HPR was defined as AUC > 46U. The primary endpoint was the rate of HPR (%) at predefined time points during the first 24 h after cangrelor cessation. Poisson regression was used to estimate the relationship between the overlap time of cangrelor and ticagrelor co-administration and the number of subsequent HPR episodes, expressed as incidence rate ratio (IRR) with 95% confidence interval (95%CI). Between December 2017 and October 2019 16 patients (81% male, 58 years) were enrolled. On average, ticagrelor was administered 39 min (IQR 5–50) before the end of cangrelor infusion. The rate of HPR was highest 90 min after cangrelor cessation and was present in 44% (7/16) of patients. The number of HPR episodes increased significantly with decreasing overlap time of cangrelor and ticagrelor co-administration (IRR 1.03, 95%CI 1.01–1.05; *p* = 0.005). In this selected cohort of hypothermic cardiac arrest survivors who received cangrelor during PCI, ticagrelor loading within the recommended time frame before cangrelor cessation resulted in a substantial amount of patients with HPR.

## 1. Introduction

Acute coronary occlusion is a leading cause of cardiac arrest requiring immediate antiplatelet treatment and percutaneous coronary intervention (PCI) [1]. In randomized trials, newer oral P2Y12 inhibitors ticagrelor and prasugrel have been proven to be superior to clopidogrel in reducing ischemic events and have become part of standard dual antiplatelet treatment in patients with acute coronary syndromes [2,3].

Hypothermia, morphine treatment and hemodynamic compromise, however, may impair oral P2Y12 inhibitor uptake and metabolism, resulting in delayed and insufficient platelet inhibition after coronary stenting [4,5,6,7]. Both cardiac arrest and absence of P2Y12 inhibition are independent risk factors for acute stent thrombosis, a potentially fatal complication following successful PCI [8,9].

Cangrelor is the first intravenous P2Y12 inhibitor available. In a meta-analysis of three large phase III trials (CHAMPION PCI, PLATFORM and PHOENIX) consisting of patients undergoing PCI, cangrelor has been shown to reduce the rate of death, myocardial infarction, ischemia-driven revascularization, or stent thrombosis at 48 h and 30 days compared with clopidogrel, without increase in major bleeding [10]. Recently, cangrelor was shown to achieve rapid-onset platelet inhibition in cardiac arrest [11]. However, caution must be exercised when switching from cangrelor to oral P2Y12 inhibitors. In accordance with cangrelor’s product characteristics, oral P2Y12 inhibitors should be administered a maximum of 30 min prior to cangrelor cessation, or immediately thereafter [12]. This transition regimen, however, might be inappropriate for resuscitated patients treated with hypothermia, as it may pose the risk of early platelet function recovery and stent thrombosis, which carries a 40% mortality at 30 days [13].

In this study, we aimed to assess the rate of high platelet reactivity (HPR) in the first 24 h after switching from cangrelor to ticagrelor in a cohort of hypothermic cardiac arrest survivors with ST-segment myocardial infarction (STEMI).

## 2. Materials and Methods

This prospective observational cohort pilot study was conducted at the critical care unit of the Emergency Department at the General Hospital of Vienna, a 2500-bed tertiary care facility in Austria, Europe. We included adult out-of-hospital cardiac arrest survivors with STEMI, who were treated with targeted temperature management (TTM, 33 ± 1 °C), underwent acute PCI and received an intravenous P2Y12 inhibition with cangrelor (30 µg/kg intravenous bolus, 4 µg/kg/min continuous infusion), followed by a 180 mg oral loading dose of crushed ticagrelor via a nasogastric tube before cangrelor cessation. Exclusion criteria comprised a history of oral P2Y12 inhibitor therapy, administration of intravenous thrombolysis or use of an extracorporeal life support device.

The primary endpoint was the rate of HPR (%) at predefined time points during the first 24 h after cangrelor cessation. Secondary endpoints were the rate of HPR at the time of stent placement and the relationship between the overlap time of cangrelor and ticagrelor co-administration (time of ticagrelor administration before cangrelor cessation) and the number of subsequent HPR episodes. The study was approved by the Ethics Committee of the Medical University of Vienna (EC# 1674/2013) and conducted in accordance with the latest version of the Helsinki declaration. A waiver for written informed consent was obtained from the Ethics Committee. The consent was permanently waived if the patient did not regain consciousness. Those who regained consciousness were informed of their participation as soon as they were able to understand the purpose of the study.

Upon study inclusion, demographics and resuscitation characteristics were recorded. Resuscitation-related parameters (initial rhythm, witness status, basic life support, downtime, the amount of epinephrine administered and the administration of heparin and aspirin by the emergency medical service) were collected via structured telephone interviews with the dispatch center, the paramedics at the scene and the bystander who made the emergency call. Platelet function was assessed at stent placement and at 30, 60, 90, 120, 240, 360 and 1440 min after cangrelor cessation. Platelet function was measured using whole blood impedance aggregometry (Multiplate Analyzer, Roche Diagnostics) and expressed as area under the curve (AUC) in units (U) [14,15]. Adenosine diphosphate (ADP, 6.4μM) was used to induce platelet aggregation (reference range: 29–118). HPR was defined as AUC > 46U as recommended [16,17]. Major bleeding [3] and stent thrombosis events [18] were documented during the 24 h study period. The neurologic function was assessed at hospital discharge using the five-point cerebral performance category scale (CPC 1/2: good neurologic function) [19].

### Statistical Methods

Categorical data are presented as absolute numbers (n) and relative frequencies (%), continuous data as medians with 25–75% interquartile range (IQR). We used exact Poisson regression to estimate the effect of the overlap time of ticagrelor and cangrelor co-administration in minutes and potential confounding co-variables on the number of subsequent HPR episodes, expressed as incidence rate ratio (IRR) with a 95% confidence interval (95%CI). Co-variables included co-administered drugs, blood pressure levels (mmHg), heart rate (bpm), left ventricular systolic function (normal, mild, moderate, severe dysfunction), as well as pro-BNP levels (pg/mL), liver enzyme (U/l) and bilirubin levels (mg/dl) at the time of transition from cangrelor to ticagrelor. Stata Statistical Software: Release 14, 2017 College Station, TX: StataCorp LLC was used for data analysis and GraphPad Prism version 8.3.0 for Windows, GraphPad Software, La Jolla California USA to draw figures. A 2-sided *p*-value of < 0.05 was considered statistically significant.

## 3. Results

Between December 2017 and October 2019 16 patients (81% male, median 58 years) were enrolled (Figure 1).

Overall, 12 patients (75%) survived with a good neurologic outcome. Table 1 shows the demographics and resuscitation-related characteristics of the study cohort.

At the time of stent placement, P2Y12 was sufficiently inhibited by cangrelor as indicated by platelet function below the HPR threshold in 100% of patients (median 18U, IQR 10–25) (Figure 2).

An oral loading dose of 180mg ticagrelor was administered a median of 39 min (IQR 5–50) prior to cangrelor cessation. HPR was highest 90 min after end of cangrelor infusion and was present in 44% (7/16) of patients (Figure 3). Co-administered drugs at the time of transition are given in Table 2.

There was a significant relationship between the overlap time of ticagrelor and cangrelor co-administration and the number of subsequent HPR episodes with an IRR of 1.03, 95%CI 1.01–1.05; *p* = 0.005 (Figure 4). The effect remained unchanged after adjustment for co-variables (Appendix A). No major bleeding or stent thrombosis occurred during the 24 h study period.

## 4. Discussion

This cohort study assessed platelet function after transition from intravenous cangrelor to oral ticagrelor in hypothermic cardiac arrest survivors undergoing PCI. While cangrelor sufficiently inhibited P2Y12 at the time of stent placement, platelet function recovered in more than 40% of all patients following the administration of ticagrelor in accordance with the recommended transition regimen.

The transition phase from intravenous to oral P2Y12 inhibitor treatment bears the risk of insufficient platelet inhibition by two different mechanisms. First, negative pharmacodynamic interactions between cangrelor and thienopyridine P2Y12 inhibitors may occur if the thienopyridine is given early during cangrelor infusion. Cangrelor’s high P2Y12 receptor affinity prevents short-lived clopidogrel and prasugrel metabolites from receptor binding, which could ultimately result in poor P2Y12 inhibition upon cangrelor cessation [20]. However, this mechanism seems unlikely because of the relatively long half-life of ticagrelor. For this study we decided to only include patients who were transitioned to ticagrelor, because switching to clopidogrel or prasugrel in cardiac arrest survivors is rarely done at our institution, which may reflect the current awareness clinicians have of possible pharmacodynamic interactions. This behavior, however, might change following the results of the recently published ISAR-REACT trial, which compared prasugrel with ticagrelor in patients with acute coronary syndrome with or without ST-segment elevation myocardial infarction. This trial found a significantly lower incidence of death, myocardial infarction, or stroke at one year among patients who received prasugrel than among those who received ticagrelor [21].

Second, platelet function may transiently recover after cangrelor cessation if the oral P2Y12 inhibitor is administered too late, or due to unexpected slow absorption in these critically ill patients. Whilst this risk may currently not be in focus during routine care of cardiac arrest survivors undergoing PCI, it may prove to be particularly pertinent for our patient population.

Transition regimens that could probably overcome both risk scenarios and achieve immediate and uninterrupted P2Y12 inhibition may involve concomitant administration of cangrelor and ticagrelor at the time of PCI scheduling, or the extension of the cangrelor infusion beyond oral P2Y12 inhibitor loading.

While a recent meta-analysis comparing different P2Y12 inhibitors in patients undergoing PCI suggests no clinical outcome benefit of cangrelor over oral P2Y12 inhibitors [22], cangrelor may be the appropriate initial P2Y12 inhibitor for hypothermic cardiac arrest survivors with STEMI. Crushed ticagrelor seems to be a reasonable P2Y12 inhibitor for early oral loading in this population, given the solid pharmacodynamic evidence for its safe and effective use in non-cardiac arrest patients receiving cangrelor [23]. Variable P2Y12 inhibitor kinetics in hypothermic cardiac arrest patients may render thienopyridines less suitable in view of their possible pharmacodynamic interactions. This is in line with a recent expert statement on switching platelet P2Y12 receptor-inhibiting therapies. Angiolillo and colleagues state that “early administration of ticagrelor should be considered over administration at the end of cangrelor infusion because it would minimize the potential gap in platelet inhibition during the transition phase” [24]. In a previous randomized trial on platelet aggregation during the transition from clopidogrel to ticagrelor in patients with acute coronary syndrome, a 180mg loading dose of ticagrelor yielded no additive effect on platelet aggregation or the onset of drug action compared to a 90mg dose of ticagrelor [25]. Early co-administration of cangrelor and ticagrelor 90mg could thus be an appropriate strategy in cardiac arrest survivors to ensure fast and uninterrupted P2Y12 inhibition along with a minimized bleeding risk during the transition phase. However, whether platelet suppression by simultaneous intravenous and oral P2Y12 inhibition after cardiac arrest may translate into better clinical outcomes, or may instead increase the risk of bleeding, needs further investigation.

### Limitations

The study is limited by its observational design and the small sample size of a highly selected patient cohort consisting of hypothermic cardiac arrest survivors with STEMI. We assessed platelet reactivity as a surrogate marker of thrombotic complications. The study was not powered for clinical outcomes. Compensation for potential confounding was attempted by adjustment for clinically plausible co-variables. Multivariable analyses including each relevant co-variable separately did not indicate relevant confounding. However, appropriate caution must be exercised when interpreting our results, which do not allow any conclusions to be drawn for clinical practice. Furthermore, the study only included patients who were transitioned from cangrelor to the oral P2Y12 inhibitor ticagrelor. From our results, no inferences must be made as to platelet reactivity or potential drug-drug interactions after transitioning from cangrelor to clopidogrel or prasugrel.

Furthermore, no gold standard assay for the assessment of platelet function is available. We used whole blood impedance aggregometry, which is a standardized platelet function assay shown to predict stent thrombosis more reliably than the vasodilator-stimulated phosphoprotein phosphorylation assay [14]. The 46U HPR threshold used in this study is a consensus-defined cut-off based on recommendations from expert opinion papers [26,27,28]. However, to date, no reference threshold for platelet function indicating a clinically relevant lack of platelet inhibition has been established. When interpreting our results, it needs to be taken into account, that platelet function assessment using different assays and/or the use of an HPR threshold other than 46U may lead to different results. A test based on ADP-induced platelet aggregation may lack the specificity to evaluate P2Y12 function in the early phase after successful resuscitation. HPR rates in our study may thus be underestimated.

## 5. Conclusions

In this observational cohort pilot study, we found a substantial rate of HPR after PCI in hypothermic cardiac arrest survivors receiving oral ticagrelor loading within the proposed time window before cangrelor cessation. Considering the pharmacological properties of available P2Y12 inhibitors, co-administration of ticagrelor with cangrelor at the time of PCI planning may be reasonable, but more research is needed. It would be pertinent to perform studies with larger sample sizes to investigate the benefit risk profile of alternative transition regimens, in order to optimize timing of P2Y12 inhibitor administration in successfully resuscitated patients undergoing coronary stenting.

## Figures and Tables

**Figure 1 jcm-09-00583-f001:**
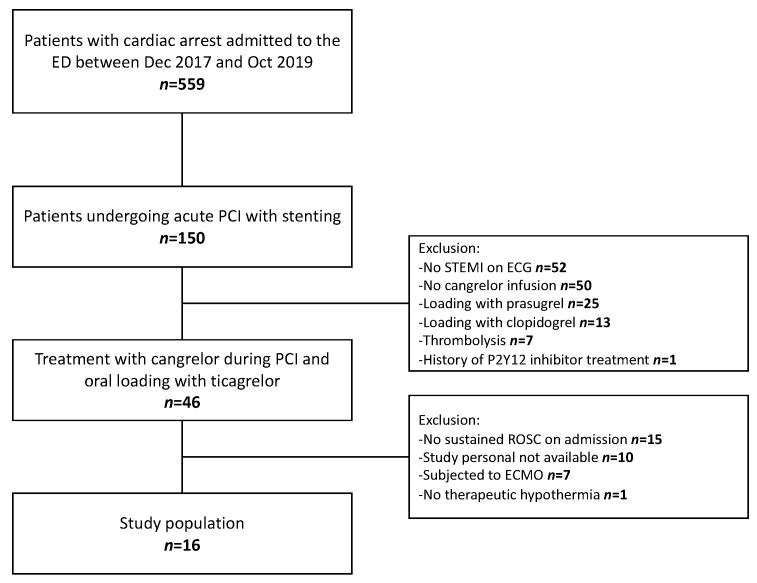
Flowchart of the study.

**Figure 2 jcm-09-00583-f002:**
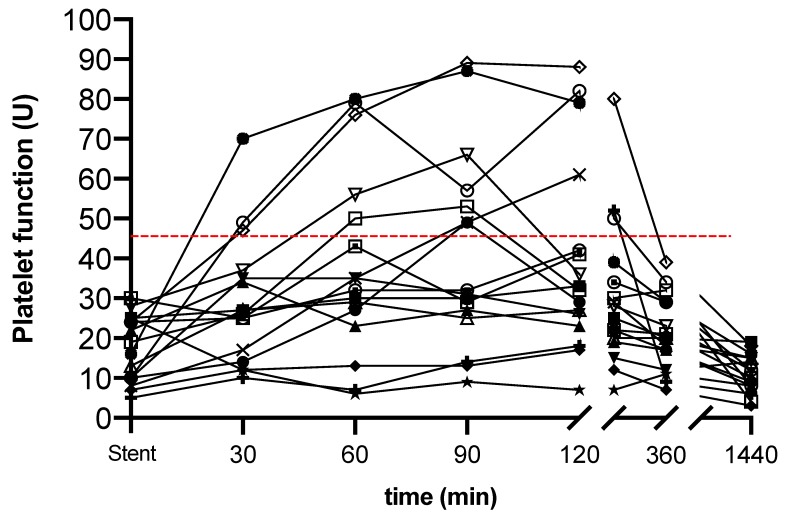
Platelet function from coronary stent placement (stent) to 24 h after end of cangrelor infusion (x-axis) was measured using whole blood aggregometry and is given in U (y-axis). Minutes 30 to 1440 refer to the time after cangrelor cessation. Cangrelor sufficiently inhibited P2Y12 at the time of coronary stent placement. Transitioning to ticagrelor within 39 min (IQR 5–50) before cangrelor cessation resulted in high platelet reactivity (HPR) in 44% (7/16) of patients within the first 90 min after end of cangrelor infusion. Red dashed line, HPR threshold of >46U.

**Figure 3 jcm-09-00583-f003:**
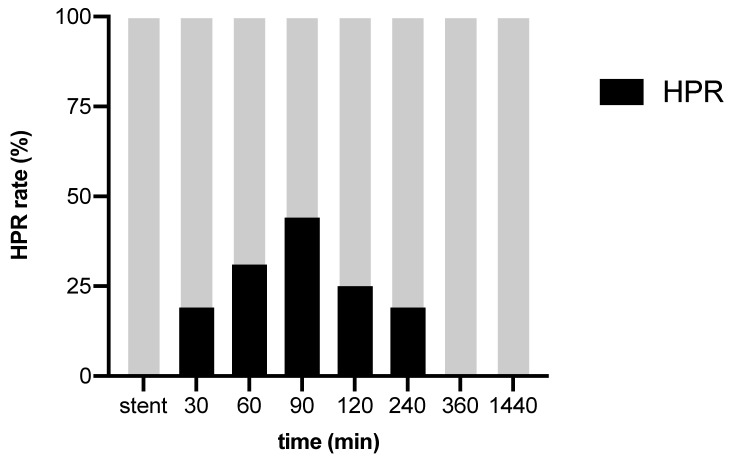
Rate of high platelet reactivity (HPR) after end of cangrelor infusion. Minutes 30 to 1440 refer to the time after cangrelor cessation. Cangrelor sufficiently inhibited P2Y12 at the time of stent placement (stent) in 100% of patients. After cangrelor cessation, the rate of HPR increased from 20% at 30 min to 44% at 90 min, and was still present in 20% of patients at 240 min.

**Figure 4 jcm-09-00583-f004:**
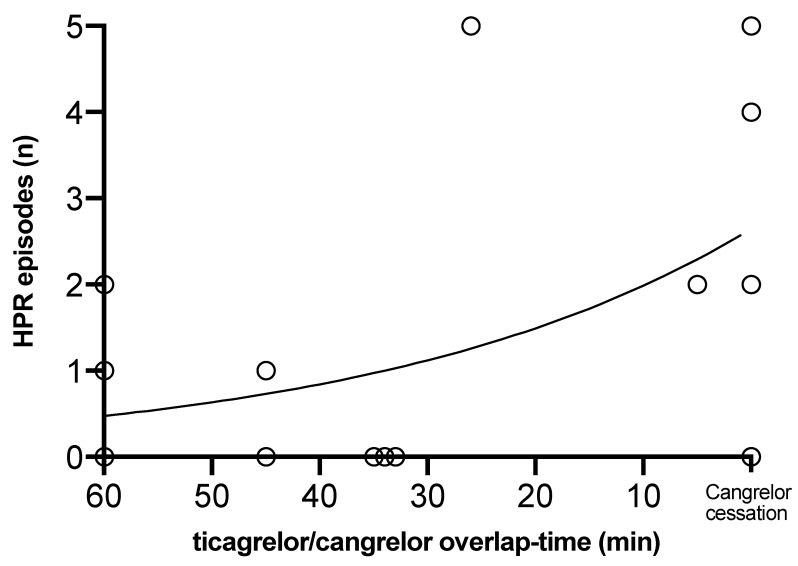
Relationship between the overlap time of cangrelor and ticagrelor co-administration (x-axis) and the number of high platelet reactivity (HPR) episodes after cangrelor cessation (y-axis). HPR episodes significantly increased with decreasing ticagrelor/cangrelor overlap time.

**Table 1 jcm-09-00583-t001:** Characteristics of study patients. Data are n (%) or median (25–75% IQR). Blood values were measured at the time of stent placement.

Variable	Total N = 16
Male sex	13 (81)
Age, years	58 (45–61)
BMI, kg/m^2^	27 (25–29)
Comorbidities	
Diabetes mellitus	2 (13)
Hypertension	3 (19)
Smoker	3 (19)
Chronic heart disease	0
Shockable rhythm	16 (100)
Witnessed	13 (81)
Basic life support	13 (81)
Epinephrine, mg	3 (2–4)
4000IE heparin ^1^	16 (100)
250 mg aspirin ^1^	16 (100)
Downtime ^2^, min	19 (14–30)
Lactate, mmol/L (1.8) ^3^	3.4 (2.3–8.7)
Troponin T, ng/L (14) ^3^	206 (110–227)
Platelet count, x10E9/L (150–350) ^3^	244 (215–358)
NT-proBNP, pg/mL (125) ^3^	236 (126–416)
ASAT, U/l (17–59; 14–36) ^4^	280 (170–646)
ALAT, U/l (50; 35) ^4^	142 (105–241)
Gamma-GT, U/L (15–73; 12–43) ^4^	69 (57–122)
Bilirubin, mg/dL (1.20) ^3^	0.53 (0.44–0.74)
Blood pressure (BP), mmHg	
- Systolic BP	112 (98–131)
- Diastolic BP	66 (59–78)
- Mean BP	81 (76–94)
Heart rate, bpm	78 (48–84)
Temperature, °C	33 (33–34)
Left ventricular systolic function	
- Normal	3 (19)
- Mild dysfunction	8 (50)
- Moderate dysfunction	2 (13)
- Severe dysfunction	3 (19)
Duration of cangrelor infusion, min	147 (127–180)
Ticagrelor administration before cangrelor cessation, min	39 (5–50)
Number of implanted coronary stents	1 (1–2)
CPC 1-2 at hospital discharge	12 (75)

^1^ All patients received 4000IU unfractionated heparin and 250 mg aspirin from the emergency medical service prior to hospital admission. ^2^ Downtime represents the interval from collapse to return of spontaneous circulation. ^3^ Upper limit of normal or reference range. ^4^ Upper limit of normal or reference range for male; female. ALAT, alanine aminotransferase; ASAT, aspartate aminotransferase; BMI, body mass index; bpm, beats per minute; CPC, cerebral performance category; Gamma-GT, Gamma-glutamyltransferase; NT-proBNP, N-terminal pro-brain natriuretic peptide.

**Table 2 jcm-09-00583-t002:** Co-administered drugs at the time of transition from cangrelor to ticagrelor. Data are n (%) and median (25 to 75% IQR).

	Number of Patients (n, %)	Dose (Median, IQR)
Continuous administration		
Norepinephrine (µg/kg/min)	12 (75)	0.061 (0.050–0.129)
Propofol 2% (mg/kg/h)	13 (81)	1.33 (1.20–1.71)
Midazolam (µg/kg/h)	3 (19)	0.211 (0.171–0.217)
Remifentanil (µg/kg/min)	13 (81)	0.106 (0.090–0.118)
Fentanyl (µg/kg/h)	3 (19)	2.000 (2.000–2.053)
Rocuronium (mg/h)	16 (100)	21.75 (18.00–25.50)
Insulin (IU/h)	2 (13)	3.5 (2.75–4.25)
Bolus administration		
Amoxicillin/Clavulanic Acid (g)	2 (13)	2.2
Pantoprazole (mg)	3 (19)	40
Amiodarone (mg)	3 (19)	300
Atorvastatin (mg)	2 (13)	80

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
