# Peer review of "High Platelet Reactivity after Transition from Cangrelor to Ticagrelor in Hypothermic Cardiac Arrest Survivors with ST-Segment Elevation Myocardial Infarction"

_jcm, 2020, doi:10.3390/jcm9020583_

Round 1

Reviewer 1 Report

In the present study, Buchtele et al report the effect of switching from the intravenous P2Y12 inhibitor cangrelor to ticagrelor, an oral P2Y12 inhibitor, in cardiac arrest survivors. The authors show that this regimen results in a greater rate of high platelet reactivity.  While the present data represents a population not so well characterised, the small sample size is a major issue. Thus, the present study is critically underpowered to warrant any reasonable conclusions to be drawn by the authors. Additionally, the follow-up period of 24 hours is rather limited. This is in contrast with previous trials, such as the PLATO trial, STEEL-PCI, and PEGASUS trial, which evaluated effects of ticagrelor over longer follow-up periods. The Introduction fails to provide convincing background review of previous key trials of cangrelor and ticagrelor.

Author Response

Authors’ response:

Thank you very much for your comments.

1)        We agree with the reviewer. The study is limited by its observational design and the small sample size of a selected patient cohort. Thus, appropriate caution must be exercised when interpreting our results. We tried to point out all critical issues in the Limitations Section. The reviewer is absolutely right that our results do not allow any conclusions for clinical practice to be drawn and that possible clinical implications of the pharmacodynamic findings of our study need further investigation in adequately powered clinical trials.

Data on platelet function in cardiac arrest patients receiving cangrelor during PCI are scarce [1,2]. As to yet no study has investigated the dynamics of platelet reactivity during and shortly after the transition phase from iv to oral P2Y12 inhibition in cardiac arrest survivors.

Although our study clearly was not powered for clinical outcomes, we found a high rate of HPR, which is a surrogate marker of thrombotic complications, using a whole blood impedance aggregometry with ADP test, a standardized, widely used platelet function assay. The 46U HPR threshold used in the study is a consensus-defined cut-off based on best available evidence and in line with recommendations found in expert opinion papers [3-6], which has been shown to be highly significantly associated with stent thrombosis in a large collaborative analysis “on the role of platelet reactivity for stent thrombosis risk stratification after PCI in patients on P2Y12-inhibitors” to determine the prognostic impact of platelet reactivity (P = 0.00001; risk ratio 2.73; 95%CI: 2.03–3.69) [7].

We did not perform a formal hypothesis test based sample size calculation, but estimated our sample size on expected precision of the estimates. Assuming an average event rate of 0.5 we calculated exact Poisson 95% confidence intervals and plotted these against sample size (exposure) [8-11]. A sample size of 16 individuals turned out to be the optimum value located at the steep and flat part of the precision vs. sample size curve (please see the uploaded pdf version of the point by point reply to see the figure)

Although no inferences for clinical practice can be made from our study results, its findings and conclusions are in line with recent expert statements on switching platelet P2Y12 receptor-inhibiting therapies. Angiolillo and colleagues suggest that “early administration of ticagrelor should be considered over administration at the end of cangrelor infusion because it would minimize the potential gap in platelet inhibition during the transition phase” [12].

References:

Fiore, M.; Gerbaud, E.; Coste, P.; Cetran, L.; Marchand, H.; Seguy, B. Optimal platelet inhibition with cangrelor in comatose survivors of out-of-hospital cardiac arrest undergoing primary percutaneous coronary intervention. Resuscitation 2018, 130, e1-e2 Pruller F, Bis L, Milke OL, Fruhwald F, Patzold S, Altmanninger-Sock S, et al. Cangrelor Induces More Potent Platelet Inhibition without Increasing Bleeding in Resuscitated Patients. Journal of clinical medicine 2018; 7. Sibbing D, Schulz S, Braun S, Morath T, Stegherr J, Mehilli J, Schomig A, von Beckerath N, Kastrati A. Antiplatelet effects of clopidogrel and bleeding in patients undergoing coronary stent placement. J Thromb Haemost 2010;8:250–256. Sibbing D, Steinhubl SR, Schulz S, Schomig A, Kastrati A. Platelet aggregation and its association with stent thrombosis and bleeding in clopidogrel-treated patients: initial evidence of a therapeutic window. J Am Coll Cardiol 2010;56:317–318. Tantry US, Bonello L, Aradi D, Price MJ, Jeong YH, Angiolillo DJ, Stone GW, Curzen N, Geisler T, Ten Berg J, Kirtane A, Siller-Matula J, Mahla E, Becker RC, Bhatt DL, Waksman R, Rao SV, Alexopoulos D, Marcucci R, Reny JL, Trenk D, Sibbing D, Gurbel PA. Consensus and update on the definition of on-treatment platelet reactivity to adenosine diphosphate associated with ischemia and bleeding. J Am Coll Cardiol 2013;62:2261–2273. Aradi D, Storey RF, Komocsi A, Trenk D, Gulba D, Kiss RG, Husted S, Bonello L, Sibbing D, Collet JP, Huber K, Working Group on Thrombosis of the European Society of C. Expert position paper on the role of platelet function testing in patients undergoing percutaneous coronary intervention. Eur Heart J 2014;35:209–215. Aradi D, Kirtane A, Bonello L, Gurbel PA, Tantry US, Huber K, et al. Bleeding and stent thrombosis on P2Y12-inhibitors: collaborative analysis on the role of platelet reactivity for risk stratification after percutaneous coronary intervention. European heart journal 2015; 36: 1762-1771. Flierl, U.; Rontgen, P.; Zauner, F.; Tongers, J.; Berliner, D.; Bauersachs, J.; Schafer, A. Platelet inhibition with prasugrel in patients with acute myocardial infarction undergoing therapeutic hypothermia after cardiopulmonary resuscitation. Thromb Haemost 2016, 115, 960-968 Franchi, F.; Rollini, F.; Rivas, A.; Wali, M.; Briceno, M.; Agarwal, M.; Shaikh, Z.; Nawaz, A.; Silva, G.; Been, L., et al. Platelet Inhibition With Cangrelor and Crushed Ticagrelor in Patients With ST-Segment-Elevation Myocardial Infarction Undergoing Primary Percutaneous Coronary Intervention. Circulation 2019, 139, 1661-1670 Parodi G, Valenti R, Bellandi B, Migliorini A, Marcucci R, Comito V, et al. Comparison of prasugrel and ticagrelor loading doses in ST-segment elevation myocardial infarction patients: RAPID (Rapid Activity of Platelet Inhibitor Drugs) primary PCI study. Journal of the American College of Cardiology 2013; 61: 1601-1606. Holm M, Tornvall P, Henareh L, Jensen U, Golster N, Alstrom P, et al. The MOVEMENT Trial. Journal of the American Heart Association 2019; 8: e010152. Angiolillo DJ, Rollini F, Storey RF, Bhatt DL, James S, Schneider DJ, et al. International Expert Consensus on Switching Platelet P2Y12 Receptor-Inhibiting Therapies. Circulation 2017; 136: 1955-1975.

2)        We agree with the reviewer that a prolonged follow-up of clinical outcome endpoints would be reasonable and should be addressed in further larger sampled randomized studies, which are designed and sufficiently powered for clinical outcomes. The focus of the current study, however, was the rate of high platelet reactivity during and shortly after the transition phase from cangrelor to ticagrelor (this happens, at the latest, 4 hours after start of cangrelor infusion according to its SPC) and all study patients, including those with HPR, showed sufficient P2Y12 inhibition at 24 hours after transition from cangrelor to ticagrelor. High platelet reactivity occurring after 24 hours is likely independent of pharmacodynamic interactions during the iv to oral P2Y12 inhibitor switching phase; its assessment, although clearly relevant for outcome trials, was beyond the scope of the current study.

To better highlight the mechanistic study design not driven by clinical endpoints, we re-phrased the description of the study design into prospective observational cohort pilot study (Methods Section).

Furthermore, it is now stated more clearly in both the Limitations Section and the Conclusions Section that our results do not allow any conclusions or implications to be drawn for clinical practice, and that adequately powered clinical studies are needed to reliably investigate the pharmacodynamic findings of our study and the clinical implications thereof.

Limitations (P8/L207-216): “The study is limited by its observational design and the small sample size of a highly selected patient cohort consisting of hypothermic cardiac arrest survivors with STEMI. We assessed platelet reactivity as a surrogate marker of thrombotic complications. The study was not powered for clinical outcomes. Compensation for potential confounding was attempted by adjustment for clinically plausible co-variables. Multivariable analyses including each relevant co-variable separately did not indicate relevant confounding. However, appropriate caution must be exercised when interpreting our results, which do not allow any conclusions to be drawn for clinical practice. Furthermore, the study only included patients who were transitioned from cangrelor to the oral P2Y12 inhibitor ticagrelor. From our results no inferences must be made as to platelet reactivity or potential drug-drug interactions after transitioning from cangrelor to clopidogrel or prasugrel.”

Conclusions (P9/L231-236): “Considering the pharmacological properties of available P2Y12 inhibitors, co-administration of ticagrelor with cangrelor at the time of PCI planning may be a reasonable approach, but more research is needed. It would be pertinent to perform studies with larger sample sizes to investigate the benefit-risk profile of different transition regimens in order to optimize timing of P2Y12 inhibitor administration in successfully resuscitated patients undergoing coronary stenting.”

3)        The Introduction Section has been improved. Key trials of cangrelor and ticagrelor have been added to the text (including appropriate references):

Introduction (P2/L50-54): “In a meta-analysis of three large phase III trials (CHAMPION PCI, PLATFORM and PHOENIX) consisting of patients undergoing PCI, cangrelor has been shown to reduce the rate of death, myocardial infarction, ischemia-driven revascularization, or stent thrombosis at 48 hours and 30 days compared with clopidogrel, without increase in major bleeding [1].”

Introduction (P1,2/L42-45): “In randomized trials newer oral P2Y12 inhibitors ticagrelor and prasugrel have been proven to be superior to clopidogrel in reducing ischemic events and have become part of standard dual antiplatelet treatment in patients with acute coronary syndromes [2,3].

References:

Majmundar M, Kansara T, Jain A, Shah P, Mithawala P, Desai R, et al. Meta-Analysis of the Role of Cangrelor for Patients Undergoing Percutaneous Coronary Intervention. The American journal of cardiology 2019; 123: 1069-1075. D. Wiviott, E. Braunwald, C.H. McCabe, et al.Prasugrel versus clopidogrel in patients with acute coronary syndromes. The New England Journal of Medicine, 357 (2007), pp. 2001-2015 Wallentin, R.C. Becker, A. Budaj, et al.Ticagrelor versus clopidogrel in patients with acute coronary syndromes. The New England Journal of Medicine, 361 (2009), pp. 1045-1057

3)        Compensation for potential confounding was attempted by adjustment for relevant co-variables (following a suggestion by Reviewer#4). We estimated the effect of these potentially confounding co-variables on the number of HPR events in exact Poisson regression models. Relevant co-variables included: co-administered medications and blood pressure levels, heart rate, temperature, left ventricular systolic function as well as pro-BNP and liver enzyme levels at the time of switching drugs. This has been added to the Methods Section. Results are now available with the supplement: The effect of the overlap-time of ticagrelor and cangrelor co-administration remained unchanged after adjustment for co-variables.

Furthermore, a table showing co-administered drugs at the time of transition was added to the manuscript (Table 2).

Reviewer 2 Report

Authors present paper about hypothermic cardiac arrest survivors, but as a Reviewer I have some comments and suggestions.  

Authors have to chcek one more time the Instruction for Authors of Journal of Clinical Medicine.

Authors have to verify the space between words in the whole manuscript (for example – line 30 in the bracket)

The abstract should be without headings (Background etc.).

Introduction is very short – provide more information.

Second section should be Materials and Methods not Experimental Section.

Authors should provide also exclusdion criteria, information about when and where was the study provide.

Supplementary Materials should be presented in main paper.

line 112 – this information should be smaller size of letter

Figure 1 –this figure need legend

Discussion as the Introduction is very short – provide more information – compare your results with other Authors’ research. 

Author Response

Reviewer #2

Authors present paper about hypothermic cardiac arrest survivors, but as a Reviewer I have some comments and suggestions.  

Authors have to chcek one more time the Instruction for Authors of Journal of Clinical Medicine.

Authors’ response: Thank you. The Journal’s Instructions for Authors have been re-checked and the manuscript has now been prepared accordingly.

Authors have to verify the space between words in the whole manuscript (for example – line 30 in the bracket)

Authors’ response: Thank you. This has been done.

The abstract should be without headings (Background etc.).

Authors’ response: This has been corrected.

Introduction is very short – provide more information.

Authors’ response: The Introduction section has been improved.

Second section should be Materials and Methods not Experimental Section.

Authors’ response: Thank you. This has been corrected.

Authors should provide also exclusdion criteria, information about when and where was the study provide. Supplementary Materials should be presented in main paper.

Authors’ response: The reviewer is right. The study was conducted at the Emergency Department at the General Hospital of Vienna, one of Europe’s largest tertiary care facilities. The Emergency Department comprises an outpatient care section and an affiliated critical care unit covering more than 90,000 patients overall and treats about 250 cardiac arrest patients per year.

Between December 2017 and October 2019 we included adult out-of-hospital cardiac arrest survivors with STEMI, who were treated with hypothermia (33±1°C), underwent acute PCI and received an intravenous P2Y12 inhibition with cangrelor, followed by a loading dose of crushed ticagrelor.

Patients with a history of oral P2Y12 inhibitor therapy and patients who received intravenous thrombolysis or were treated using an extracorporeal life support device were excluded.

This is now stated more clearly in the manuscript:

Methods (P2/L65-72): “This prospective observational cohort pilot study was conducted at the critical care unit of the Emergency Department at the general hospital of Vienna, a 2500-bed tertiary care facility in Austria, Europe. We included adult out-of-hospital cardiac arrest survivors with STEMI, who were treated with targeted temperature management (TTM, 33±1°C), underwent acute PCI and received an intravenous P2Y12 inhibition with cangrelor (30µg/kg intravenous bolus, 4µg/kg/min continuous infusion), followed by a 180mg oral loading dose of crushed ticagrelor via a nasogastric tube before cangrelor cessation. Exclusion criteria comprised a history of oral P2Y12 inhibitor therapy, administration of intravenous thrombolysis or use of an extracorporeal life support device.”

Following your suggestion, we added supplemental figure 1 (showing the study flow chart) to the main paper (Figure 1).

line 112 – this information should be smaller size of letter

Authors’ response: This has been corrected.

Figure 1 –this figure need legend

Authors’ response: Thank you.

Following legend is provided:

Platelet function from coronary stent placement (stent) to 24 hours after end of cangrelor infusion (x-axis) was measured using whole blood aggregometry and is given in U (y-axis). Minutes 30 to 1440 refer to the time after cangrelor cessation. Cangrelor sufficiently inhibited P2Y12 at the time of coronary stent placement. Transitioning to ticagrelor within 39 minutes (IQR 5-50) before cangrelor cessation resulted in high platelet reactivity (HPR) in 44% (7/16) of patients within the first 90 minutes after end of cangrelor infusion. Red dashed line, HPR threshold of >46U.”

(NB: Following your suggestion, supplemental figure 1 was added to the main text. Consequently, formerly Figure 1 has been changed into Figure 2.)

Discussion as the Introduction is very short – provide more information – compare your results with other Authors’ research.

Authors’ response: Thank you very much for your suggestions. We agree with the reviewer.

The Discussion section has been improved.

Data on platelet function in cardiac arrest patients receiving cangrelor during PCI are scarce. So far, two observational studies have investigated the antiplatelet effect of intravenous cangrelor in resuscitated patients treated with therapeutic hypothermia [1,2]. As of yet no study has investigated the dynamics of platelet reactivity during and shortly after the transition phase from iv to oral P2Y12 inhibition in cardiac arrest survivors. The limited availability of data makes it difficult to compare our results with those of previous studies.

Given the negative pharmacodynamic interactions between cangrelor and thienopyridines [3,4] on the one hand, and the safe and effective use of ticagrelor in non-cardiac arrest patients receiving cangrelor [5] on the other hand, a cangrelor-to-ticagrelor transition regimen seems reasonable in hypothermic cardiac arrest patients. The optimal timing of P2Y12 inhibitor transition, however, remains unclear. The results of our study may highlight this knowledge gap and underline the need for further research aimed at optimal timing of P2Y12 inhibitor transition. Our findings and conclusions are in line with the most recent conference expert statement on switching platelet P2Y12 receptor-inhibiting therapies. Angiolillo and colleagues state that “early administration of ticagrelor should be considered over administration at the end of cangrelor infusion because it would minimize the potential gap in platelet inhibition during the transition phase” [4].

There is solid evidence that opiate treatment, hypothermia, hemodynamic compromise and primary PCI impair the pharmacodynamic onset of oral antiplatelet agents [6-8], all of which are accentuated in cardiac arrest patients. Previous prospective cohort studies on hypothermic cardiac arrest patients with acute coronary syndrome receiving ticagrelor or prasugrel found a significant delay of sufficient P2Y12 inhibition with HPR rates of up to 40-50% four to six hours after loading [6], which compares well to the 44% HPR rate found in our study. Even in non-resuscitated patients with acute coronary syndrome treated with morphine and ticagrelor, the prevalence of high on‐treatment platelet reactivity was still 50% 2 hours after ticagrelor administration in a recent multicenter randomized clinical trial [9]. These data (along with results from our study) suggest that early concomitant administration of cangrelor and ticagrelor at the time of PCI scheduling or the extension of the cangrelor infusion beyond ticagrelor loading could possibly be appropriate transition regimens to avoid absence of P2Y12 inhibition or platelet function recovery and thus reduce the risk of stent thrombosis in cardiac arrest survivors. However, further larger-scaled research is needed before any conclusions can be drawn for clinical practice.

Discussion section (P8/L199-203): …”This in line with a recent expert statement on switching platelet P2Y12 receptor-inhibiting therapies. Angiolillo and colleagues state that “early administration of ticagrelor should be considered over administration at the end of cangrelor infusion because it would minimize the potential gap in platelet inhibition during the transition phase” [3].

As the results of the recently published ISAR-REACT trial [10] may have substantial impact on the antithrombotic treatment of patients with acute coronary syndrome undergoing PCI, including those with cardiac arrest, a more detailed description of its findings have been added to the manuscript.

Discussion section (P7,8/L180-184): … “This behavior, however, might change following the results of the recently published ISAR-REACT trial, which compared prasugrel with ticagrelor in patients with acute coronary syndrome with or without ST-segment elevation myocardial infarction. This trial found a significantly lower incidence of death, myocardial infarction, or stroke at one year among patients who received prasugrel than among those who received ticagrelor.”

Limitations section (P8/L213-216): “Furthermore, the study only included patients who were transitioned from cangrelor to the oral P2Y12 inhibitor ticagrelor. From our results no inferences must be made as to platelet reactivity or potential drug-drug interactions after transitioning from cangrelor to clopidogrel or prasugrel.”

References:

Fiore, M.; Gerbaud, E.; Coste, P.; Cetran, L.; Marchand, H.; Seguy, B. Optimal platelet inhibition with cangrelor in comatose survivors of out-of-hospital cardiac arrest undergoing primary percutaneous coronary intervention. Resuscitation 2018, 130, e1-e2 Pruller F, Bis L, Milke OL, Fruhwald F, Patzold S, Altmanninger-Sock S, et al. Cangrelor Induces More Potent Platelet Inhibition without Increasing Bleeding in Resuscitated Patients. Journal of clinical medicine 2018; 7. Schneider, D.J. Transition strategies from cangrelor to oral platelet P2Y12 receptor antagonists. Coron Artery Dis 2016, 27, 65-69 Angiolillo DJ, Rollini F, Storey RF, Bhatt DL, James S, Schneider DJ, et al. International Expert Consensus on Switching Platelet P2Y12 Receptor-Inhibiting Therapies. Circulation 2017; 136: 1955-1975. Franchi, F.; Rollini, F.; Rivas, A.; Wali, M.; Briceno, M.; Agarwal, M.; Shaikh, Z.; Nawaz, A.; Silva, G.; Been, L., et al. Platelet Inhibition With Cangrelor and Crushed Ticagrelor in Patients With ST-Segment-Elevation Myocardial Infarction Undergoing Primary Percutaneous Coronary Intervention. Circulation 2019, 139, 1661-1670 Flierl, U.; Rontgen, P.; Zauner, F.; Tongers, J.; Berliner, D.; Bauersachs, J.; Schafer, A. Platelet inhibition with prasugrel in patients with acute myocardial infarction undergoing therapeutic hypothermia after cardiopulmonary resuscitation. Thromb Haemost 2016, 115, 960-968 Ibrahim, K.; Christoph, M.; Schmeinck, S.; Schmieder, K.; Steiding, K.; Schoener, L.; Pfluecke, C.; Quick, S.; Mues, C.; Jellinghaus, S., et al. High rates of prasugrel and ticagrelor non-responder in patients treated with therapeutic hypothermia after cardiac arrest. Resuscitation 2014, 85, 649-656 Uminska, J.M.; Ratajczak, J.; Buszko, K.; Sobczak, P.; Sroka, W.; Marszall, M.P.; Adamski, P.; Steblovnik, K.; Noc, M.; Kubica, J. Impact of mild therapeutic hypothermia on bioavailability of ticagrelor in patients with acute myocardial infarction after out-of-hospital cardiac arrest. Cardiol J 2019 Holm M, Tornvall P, Henareh L, Jensen U, Golster N, Alstrom P, et al. The MOVEMENT Trial. Journal of the American Heart Association 2019; 8: e010152. Schupke, S.; Neumann, F.J.; Menichelli, M.; Mayer, K.; Bernlochner, I.; Wohrle, J.; Richardt, G.; Liebetrau, C.; Witzenbichler, B.; Antoniucci, D., et al. Ticagrelor or Prasugrel in Patients with Acute Coronary Syndromes. N Engl J Med 2019, 381, 1524-1534

Reviewer 3 Report

The manuscript by Buchtele and co-authors entitled “High platelet reactivity after transition from cangrelor to ticagrelor in hypothermic cardiac arrest survivors with ST-segment elevation myocardial infarction” deals with the question if transition from intravenous cangrelor to oral ticagrelor after percutaneous coronary intervention carries the risk of platelet function recovery which may lead to a higher risk of consecutive acute stent thrombosis in a defined cohort. 16 Patients were included in this prospective observational cohort study. The authors demonstrate that there was a substantial amount of patients (7/16 = 44%) who had high platelet reactivity (HPR) when ticagrelor loading dose was administered within the recommended time frame. According to the authors, the rate of HPR was highest 90 minutes after cangrelor cessation and the number of HPR episodes increased significantly with decreasing overlap-time of cangrelor and ticagrelor co-administration.

Comments to the authors:

Although this trial included a highly selected patient cohort (as stated by the authors in the limitations section), the sample size is too small. More patients should be recruited and authors should provide an evidence-based sample size calculation. In terms of the primary endpoint of this trial, it seems a little problematic that there is no established reference threshold. It is a further weakness of the study that no clinical outcome endpoints were included. To summarize points 1.-3., the combination of small sample size, an error-prone primary endpoint and no data with regard to clinical outcome makes interpretation of the results very difficult, especially because these findings may be highly relevant for patients. page 6, lines 163ff: The authors should briefly illustrate the findings of the ISAR-REACT trial and explain in what terms prasugrel was superior to ticagrelor. This aspect should also be added to the limitations section.

Author Response

Comments to the authors:

Although this trial included a highly selected patient cohort (as stated by the authors in the limitations section), the sample size is too small. More patients should be recruited and authors should provide an evidence-based sample size calculation. In terms of the primary endpoint of this trial, it seems a little problematic that there is no established reference threshold. It is a further weakness of the study that no clinical outcome endpoints were included. To summarize points 1.-3., the combination of small sample size, an error-prone primary endpoint and no data with regard to clinical outcome makes interpretation of the results very difficult, especially because these findings may be highly relevant for patients.

Authors’ response: Thank you very much for your comments and suggestions.

1)        We agree with the reviewer. The study is limited by its observational design and the small sample size of a selected patient cohort. Thus, appropriate caution must be exercised when interpreting our results. The reviewer is absolutely right that our results allow no conclusions to be drawn for clinical practice and that possible clinical implications of the pharmacodynamic findings of our study need further investigation in adequately powered clinical trials. We tried to point out all critical issues in the Limitations Section.

However, it is now stated more clearly in both the Limitations Section and the Conclusions Section that the results of our study do not allow us to draw any conclusions for clinical practice, and that adequately powered clinical studies are needed to reliably investigate the pharmacodynamic findings of our study and the clinical implications thereof.

Limitations (P8/L207-216): “The study is limited by its observational design and the small sample size of a highly selected patient cohort consisting of hypothermic cardiac arrest survivors with STEMI. We assessed platelet reactivity as a surrogate marker of thrombotic complications. The study was not powered for clinical outcomes. Compensation for potential confounding was attempted by adjustment for clinically plausible co-variables. Multivariable analyses including each relevant co-variable separately did not indicate relevant confounding. However, appropriate caution must be exercised when interpreting our results, which do not allow any conclusions to be drawn for clinical practice. Furthermore, the study only included patients who were transitioned from cangrelor to the oral P2Y12 inhibitor ticagrelor. From our results no inferences must be made as to platelet reactivity or potential drug-drug interactions after transitioning from cangrelor to clopidogrel or prasugrel”

Conclusions (P9/L231-236): “Considering the pharmacological properties of available P2Y12 inhibitors, co-administration of ticagrelor with cangrelor at the time of PCI planning may be a reasonable approach, but more research is needed. It would be pertinent to perform studies with larger sample sizes to investigate the benefit-risk profile of different transition regimens in order to optimize timing of P2Y12 inhibitor administration in successfully resuscitated patients undergoing coronary stenting”.

To better highlight the mechanistic study design not driven by clinical endpoints, we re-phrased the description of the study design into prospective observational cohort pilot study (Methods Section)

2)        We further agree with the reviewer that a prolonged follow-up of clinical outcome endpoints would be reasonable and should be addressed in further larger sampled randomized studies, which are designed and adequately powered to reliably assess clinically relevant endpoints. The focus of the current study, however, was the rate of HPR during and after the transition phase from cangrelor to ticagrelor (which happens at latest 4 hours after start of cangrelor infusion according to its SPC) and all of our study patients, including those with HPR, showed sufficient P2Y12 inhibition at 24 hours after switching from cangrelor to ticagrelor. High platelet reactivity occurring after 24 hours are likely independent of pharmacodynamic interactions during the drug switching phase, but may indicate severely delayed onset of action [1-3]; its assessment, although clearly highly relevant in large-scaled outcome trials, was beyond the scope of the current study.

No data are available yet on platelet reactivity during or shortly after the transition from iv to oral P2Y12 inhibitors in successfully resuscitated patients treated with hypothermia. Established transition regimens given in the SPCs have not yet been investigated in the vulnerable population of cardiac arrest survivors, who are prone to experiencing acute stent thrombosis after PCI in the absence of platelet inhibition [4-5]. Opiate treatment, hypothermia, hemodynamic compromise and primary PCI are settings known to reduce the pharmacodynamic onset of oral antiplatelet agents [6-10], all of which are accentuated in cardiac arrest patients.

Given the lack of data on how to switch P2Y12-inhibiting therapies in cardiac arrest survivors, clinicians are left with limited guidance.  The results of our study may highlight this knowledge gap and underline the need for further research aimed at optimal timing of P2Y12 inhibitor transition in successfully resuscitated patients. Despite the limitations of our study, our findings and conclusions are in line with a recent expert statement on switching platelet P2Y12 receptor-inhibiting therapies. Angiolillo and colleagues state that  “early administration of ticagrelor should be considered over administration at the end of cangrelor infusion because it would minimize the potential gap in platelet inhibition during the transition phase” [11]. Nonetheless, we definitely agree with the reviewer that our results do not allow any recommendations to be made for clinicians, and that the pharmacodynamic findings of our study need further investigation in adequately powered clinical trials.

References:

Flierl, U.; Rontgen, P.; Zauner, F.; Tongers, J.; Berliner, D.; Bauersachs, J.; Schafer, A. Platelet inhibition with prasugrel in patients with acute myocardial infarction undergoing therapeutic hypothermia after cardiopulmonary resuscitation. Thromb Haemost 2016, 115, 960-968 Ibrahim, K.; Christoph, M.; Schmeinck, S.; Schmieder, K.; Steiding, K.; Schoener, L.; Pfluecke, C.; Quick, S.; Mues, C.; Jellinghaus, S., et al. High rates of prasugrel and ticagrelor non-responder in patients treated with therapeutic hypothermia after cardiac arrest. Resuscitation 2014, 85, 649-656 Uminska, J.M.; Ratajczak, J.; Buszko, K.; Sobczak, P.; Sroka, W.; Marszall, M.P.; Adamski, P.; Steblovnik, K.; Noc, M.; Kubica, J. Impact of mild therapeutic hypothermia on bioavailability of ticagrelor in patients with acute myocardial infarction after out-of-hospital cardiac arrest. Cardiol J 2019 Joffre, J.; Varenne, O.; Bougouin, W.; Rosencher, J.; Mira, J.P.; Cariou, A. Stent thrombosis: an increased adverse event after angioplasty following resuscitated cardiac arrest. Resuscitation 2014, 85, 769-773 Penela, D.; Magaldi, M.; Fontanals, J.; Martin, V.; Regueiro, A.; Ortiz, J.T.; Bosch, X.; Sabate, M.; Heras, M. Hypothermia in acute coronary syndrome: brain salvage versus stent thrombosis? J Am Coll Cardiol 2013, 61, 686-687 Thomas MR, Morton AC, Hossain R, Chen B, Luo L, Shahari NN, Hua P, Beniston RG, Judge HM, Storey RF. Morphine delays the onset of action of prasugrel in patients with prior history of ST-elevation myocardial infarction. Thromb Haemost. 2016;116:96–102. doi: 10.1160/TH16-02-0102. Silvain J, Storey RF, Cayla G, Esteve JB, Dillinger JG, Rousseau H, Tsatsaris A, Baradat C, Salhi N, Hamm CW, Lapostolle F, Lassen JF, Collet JP, Ten Berg JM, Van’t Hof AW, Montalescot G. P2Y12 receptor inhibition and effect of morphine in patients undergoing primary PCI for ST-segment

elevation myocardial infarction: the PRIVATE-ATLANTIC study. Thromb Haemost. 2016;116:369–378. doi: 10.1160/TH15-12-0944.

Kubica J, Adamski P, Ostrowska M, Sikora J, Kubica JM, Sroka WD, Stankowska K, Buszko K, Navarese EP, Jilma B, Siller-Matula JM, Marszałł MP, Rość D, Koziński M. Morphine delays and attenuates ticagrelor exposure and action in patients with myocardial infarction: the randomized,

double-blind, placebo-controlled IMPRESSION trial. Eur Heart J. 2016;37:245–252. doi: 10.1093/eurheartj/ehv547.

Parodi G, Bellandi B, Xanthopoulou I, Capranzano P, Capodanno D, Valenti R, Stavrou K, Migliorini A, Antoniucci D, Tamburino C, Alexopoulos D. Morphine is associated with a delayed activity of oral antiplatelet agents in patients with ST-elevation acute myocardial infarction undergoing primary percutaneous coronary intervention. Circ Cardiovasc Interv. 2015;8: e001593. doi: 10.1161/CIRCINTERVENTIONS.114.001593. Franchi F, Rollini F, Angiolillo DJ. Antithrombotic therapy for patients with STEMI undergoing primary PCI. Nat Rev Cardiol. 2017;14:361–379. doi: 10.1038/nrcardio.2017.18. Angiolillo DJ, Rollini F, Storey RF, Bhatt DL, James S, Schneider DJ, et al. International Expert Consensus on Switching Platelet P2Y12 Receptor-Inhibiting Therapies. Circulation 2017; 136: 1955-1975.

3)        We used whole blood impedance aggregometry with ADP test to determine ADP-dependent platelet function, which is a standardized, widely used platelet function assay [1-2, 5]. The 46U HPR threshold is a consensus-defined cut-off based on best available evidence and in line with recommendations found in expert opinion papers [3-4]. A 46U HPR threshold was likewise chosen in the large collaborative analysis “on the role of platelet reactivity for stent thrombosis risk stratification after PCI in patients on P2Y12-inhibitors” to determine the prognostic impact of platelet reactivity. The 46U HPR cut-off was highly significantly associated with stent thrombosis (P = 0.00001; risk ratio 2.73; 95%CI: 2.03–3.69) [5].

The reviewer is certainly right, however, that the use of different platelet function assays or a different threshold may lead to different results and that there is a need to validate clinically relevant cut-off values for HPR to find the optimal range of platelet reactivity, thus enabling effective and safe antiplatelet therapy. This limitation, however, applies to all platelet function assays available and is inherent to all previous studies, which have investigated platelet function. We tried to point this critical issue out in the Limitations Section.

This issue is now more clearly stated in the Limitations Section (including appropriate references)

Limitations (P8/L217-225): …“Furthermore, no gold standard assay for the assessment of platelet function is available. We used whole blood impedance aggregometry, which is a standardized platelet function assay shown to predict stent thrombosis more reliably than the vasodilator-stimulated phosphoprotein phosphorylation assay [11]. The 46U HPR threshold used in this study is a consensus-defined cut-off based on recommendations from expert opinion papers [1-3]. However, to date, no reference threshold for platelet function indicating a clinically relevant lack of platelet inhibition has been established. When interpreting our results, it needs to be taken into account, that platelet function assessment using different assays and/or the use of an HPR threshold other than 46U may lead to different results.”

References:

Tantry US, Bonello L, Aradi D, Price MJ, Jeong YH, Angiolillo DJ, Stone GW, Curzen N, Geisler T, Ten Berg J, Kirtane A, Siller-Matula J, Mahla E, Becker RC, Bhatt DL, Waksman R, Rao SV, Alexopoulos D, Marcucci R, Reny JL, Trenk D, Sibbing D, Gurbel PA. Consensus and update on the definition of on-treatment platelet reactivity to adenosine diphosphate associated with ischemia and bleeding. J Am Coll Cardiol 2013;62:2261–2273. Aradi D, Storey RF, Komocsi A, Trenk D, Gulba D, Kiss RG, Husted S, Bonello L, Sibbing D, Collet JP, Huber K, Working Group on Thrombosis of the European Society of C. Expert position paper on the role of platelet function testing in patients undergoing percutaneous coronary intervention. Eur Heart J 2014;35:209–215. Aradi D, Kirtane A, Bonello L, Gurbel PA, Tantry US, Huber K, et al. Bleeding and stent thrombosis on P2Y12-inhibitors: collaborative analysis on the role of platelet reactivity for risk stratification after percutaneous coronary intervention. European heart journal 2015; 36: 1762-1771.

4)        Sample size calculation

Authors’ response: We did not perform a formal hypothesis test based sample size calculation, but estimated our sample size on expected precision of the estimates. Assuming an average event rate of 0.5 we calculated exact Poisson 95% confidence intervals and plotted these against sample size (exposure) [1-4]. A sample size of 16 individuals turned out to be the optimum value located at the steep and flat part of the precision vs. sample size curve (please see the attached pdf file to see the figure)

References:

Flierl, U.; Rontgen, P.; Zauner, F.; Tongers, J.; Berliner, D.; Bauersachs, J.; Schafer, A. Platelet inhibition with prasugrel in patients with acute myocardial infarction undergoing therapeutic hypothermia after cardiopulmonary resuscitation. Thromb Haemost 2016, 115, 960-968 Franchi, F.; Rollini, F.; Rivas, A.; Wali, M.; Briceno, M.; Agarwal, M.; Shaikh, Z.; Nawaz, A.; Silva, G.; Been, L., et al. Platelet Inhibition With Cangrelor and Crushed Ticagrelor in Patients With ST-Segment-Elevation Myocardial Infarction Undergoing Primary Percutaneous Coronary Intervention. Circulation 2019, 139, 1661-1670 Parodi G, Valenti R, Bellandi B, Migliorini A, Marcucci R, Comito V, et al. Comparison of prasugrel and ticagrelor loading doses in ST-segment elevation myocardial infarction patients: RAPID (Rapid Activity of Platelet Inhibitor Drugs) primary PCI study. Journal of the American College of Cardiology 2013; 61: 1601-1606. Holm M, Tornvall P, Henareh L, Jensen U, Golster N, Alstrom P, et al. The MOVEMENT Trial. Journal of the American Heart Association 2019; 8: e010152.

5)        Compensation for potential confounding was attempted by adjustment for relevant co-variables (following a suggestion by Reviewer#4). We estimated the effect of these potentially confounding co-variables on the number of HPR events in exact Poisson regression models. Relevant co-variables included: co-administered medications and blood pressure levels, heart rate, temperature, left ventricular systolic function as well as pro-BNP and liver enzyme levels at the time of switching drugs. The co-variables had no effect on the number of HPR events: Results are now available with the supplement.

Finally, a table showing co-administered drugs at the time of transition was added to the manuscript (Table 2).

page 6, lines 163ff: The authors should briefly illustrate the findings of the ISAR-REACT trial and explain in what terms prasugrel was superior to ticagrelor. This aspect should also be added to the limitations section.

Authors’ response: Thank you for this suggestion. Findings of the recent ISAR-REACT trial are now described in more detail.

Discussion Section (P7,8/L180-184): …“This behavior, however, might change following the results of the recently published ISAR-REACT trial, which compared prasugrel with ticagrelor in patients with acute coronary syndrome with or without ST-segment elevation myocardial infarction. This trial found a significantly lower incidence of death, myocardial infarction, and stroke at one year among patients who received prasugrel than among those who received ticagrelor [19].”

Limitations Section (P8/L213-216): …“Furthermore, this study only included patients who were transitioned from cangrelor to the oral P2Y12 inhibitor ticagrelor. From our results no conclusions must be drawn as to platelet reactivity or potential drug-drug interactions after transition from cangrelor to clopidogrel or prasugrel.”

Reviewer 4 Report

The authors described all pitfalls of the study in the part Limitations.

The structure of this brief, interesting, paper is according to the journal's guidelines.

It would be nice to have presented in the Table 1: systolic and diastolic pressure at the time of switching drugs, heart rate, BNP level, LVEF, liver enzymes AST, ALT GGT, bilirubin.

Co-administered drugs, not only cardiovascular drugs but antibiotics, etc. should be presented in a separate table. They could also affect the HPR.

All relevant factors should be investigated in the logistic regression analysis.  

Author Response

Reviewer #4

The authors described all pitfalls of the study in the part Limitations.

The structure of this brief, interesting, paper is according to the journal's guidelines.

It would be nice to have presented in the Table 1: systolic and diastolic pressure at the time of switching drugs, heart rate, BNP level, LVEF, liver enzymes AST, ALT GGT, bilirubin.

Co-administered drugs, not only cardiovascular drugs but antibiotics, etc. should be presented in a separate table. They could also affect the HPR.

Authors’ response: Thank you very much for your positive comments and suggestions. We greatly appreciate it.

Co-administered medications and blood pressure levels (mmHg), heart rate (bpm), temperature (°C), left ventricular systolic function (normal, mild, moderate, severe dysfunction) as well as pro-BNP (pg/ml) and liver enzyme levels at the time of switching drugs have been added to table1. Co-administered drugs at the time of transition are now presented in Table 2.

Table 2.  Co-administered drugs at the time of transition from cangrelor to ticagrelor

Number of patients (n, %)

Dose (median, IQR)

Continuous administration1

Norepinephrine (µg/kg/min)

12 (75)

0.061 (0.050-0.129)

Propofol 2% (mg/kg/h)

13 (81)

1.33 (1.20-1.71)

Midazolam (µg/kg/h)

3 (19)

0.211 (0.171-0.217)

Remifentanil (µg/kg/min)

13 (81)

0.106 (0.090-0.118)

Fentanyl (µg/kg/h)

3 (19)

2.000 (2.000-2.053)

Rocuronium (mg/h)

16 (100)

21.75 (18.00-25.50)

Insulin (IU/h)

2 (13)

3.5 (2.75-4.25)

Bolus administration

Amoxicillin/Clavulanic Acid (g)

2 (13)

2.2

Pantoprazole (mg)

3 (19)

40

Amiodarone (mg)

3 (19)

300

Atorvastatin (mg)

2 (13)

80

1Data are n (%) and median (25-75% IQR).

All relevant factors should be investigated in the logistic regression analysis.

Authors’ response:

Following your suggestion, we estimated the effect of potentially confounding co-variables on the number of HPR events using Poisson regression models including each relevant co-variable separately.

The co-variables had no effect on the number of HPR events: Results are now available with the supplement (sTable):

Exact Poisson regression

                                                         Number of obs = 16

---------------------------------------------------------------------------

hprtimepoi~s |        IRR       Suff.  2*Pr(Suff.)     [95% Conf. Interval]

-------------+-------------------------------------------------------------

timeticabe~g |   .9717907         365      0.0054       .950613    .9918672

   1.0282093         365      0.0054      1.0081328  1.049387  

---------------------------------------------------------------------------

---------------------------------------------------------------------------

hprtimepoi~s |        IRR       Suff.  2*Pr(Suff.)     [95% Conf. Interval]

-------------+-------------------------------------------------------------

timeticabe~g |   .9685399         365      0.0043      .9458258    .9903346

       m_bnp |   1.421757          10      0.5966      .5075964    3.884677

---------------------------------------------------------------------------

---------------------------------------------------------------------------

hprtimepoi~s |        IRR       Suff.  2*Pr(Suff.)     [95% Conf. Interval]

-------------+-------------------------------------------------------------

timeticabe~g |   .9782004         365      0.0499      .9556211    .9999894

        asat |   1.000424       20334      0.0775      .9999498    1.000877

---------------------------------------------------------------------------

---------------------------------------------------------------------------

hprtimepoi~s |        IRR       Suff.  2*Pr(Suff.)     [95% Conf. Interval]

-------------+-------------------------------------------------------------

timeticabe~g |   .9749758         365      0.0265      .9517738    .9971864

        alat |   1.000473       13881      0.1000      .9998996    1.001018

---------------------------------------------------------------------------

---------------------------------------------------------------------------

hprtimepoi~s |        IRR       Suff.  2*Pr(Suff.)     [95% Conf. Interval]

-------------+-------------------------------------------------------------

timeticabe~g |   .9640086         365      0.0010      .9408023    .9859071

         ggt |   1.016386        2400      0.0032      1.005273     1.02861

---------------------------------------------------------------------------

---------------------------------------------------------------------------

hprtimepoi~s |        IRR       Suff.  2*Pr(Suff.)     [95% Conf. Interval]

-------------+-------------------------------------------------------------

timeticabe~g |   .9700878         365      0.0079      .9466079      .99235

        bili |   .1565579       11.02      0.0419      .0206956    .9437993

---------------------------------------------------------------------------

---------------------------------------------------------------------------

hprtimepoi~s |        IRR       Suff.  2*Pr(Suff.)     [95% Conf. Interval]

-------------+-------------------------------------------------------------

timeticabe~g |    .975843         365      0.0155      .9548611    .9955729

lvef_red_m~h |   2.763996          13      0.0340      1.072052    7.618956

---------------------------------------------------------------------------

---------------------------------------------------------------------------

hprtimepoi~s |        IRR       Suff.  2*Pr(Suff.)     [95% Conf. Interval]

-------------+-------------------------------------------------------------

timeticabe~g |   .9703227         365      0.0046      .9487045    .9910101

   Heartrate |   .9921897        1538      0.4986      .9699581    1.015488

---------------------------------------------------------------------------

hprtimepoi~s |        IRR       Suff.  2*Pr(Suff.)     [95% Conf. Interval]

-------------+-------------------------------------------------------------

timeticabe~g |   .9720851         365      0.0060      .9507367    .9922357

       bpsys |   .9971348        2461      0.7736       .978147    1.015937

---------------------------------------------------------------------------

---------------------------------------------------------------------------

hprtimepoi~s |        IRR       Suff.  2*Pr(Suff.)     [95% Conf. Interval]

-------------+-------------------------------------------------------------

timeticabe~g |   .9755029         365      0.0180      .9534701    .9959993

       bpdia |   .9847304        1385      0.3571      .9528364    1.017429

---------------------------------------------------------------------------

---------------------------------------------------------------------------

hprtimepoi~s |        IRR       Suff.  2*Pr(Suff.)     [95% Conf. Interval]

-------------+-------------------------------------------------------------

timeticabe~g |   .9725562         365      0.0090      .9506217     .993358

      bpmean |   .9959471        1769      0.7977       .965419    1.028282

---------------------------------------------------------------------------

---------------------------------------------------------------------------

hprtimepoi~s |        IRR       Suff.  2*Pr(Suff.)     [95% Conf. Interval]

-------------+-------------------------------------------------------------

timeticabe~g |   .9693279         365      0.0030      .9477922    .9898265

           t |   .6030332       728.6      0.1125      .3126586    1.118555

---------------------------------------------------------------------------

---------------------------------------------------------------------------

hprtimepoi~s |        IRR       Suff.  2*Pr(Suff.)     [95% Conf. Interval]

-------------+-------------------------------------------------------------

timeticabe~g |   .9754559         365      0.0149      .9540476    .9953878

noradrenalin |    8.92275       3.376      0.1091      .5845038    111.3444

---------------------------------------------------------------------------

---------------------------------------------------------------------------

hprtimepoi~s |        IRR       Suff.  2*Pr(Suff.)     [95% Conf. Interval]

-------------+-------------------------------------------------------------

timeticabe~g |   .9905771         365      0.4603      .9655298    1.014458

         esm |   .8825178         389      0.0167      .7824801     .979794

---------------------------------------------------------------------------

---------------------------------------------------------------------------

hprtimepoi~s |        IRR       Suff.  2*Pr(Suff.)     [95% Conf. Interval]

-------------+-------------------------------------------------------------

timeticabe~g |   .9667655         365      0.0014      .9474538    .9865202

insulin50i~h |   1.311264          14      0.1349      .9060454    1.809148

---------------------------------------------------------------------------

---------------------------------------------------------------------------

hprtimepoi~s |        IRR       Suff.  2*Pr(Suff.)     [95% Conf. Interval]

-------------+-------------------------------------------------------------

timeticabe~g |   .9721758         365      0.0056      .9511526    .9920569

ultiva_pro~0 |   1.448134          19      0.8040      .4227409    7.927062

---------------------------------------------------------------------------

---------------------------------------------------------------------------

hprtimepoi~s |        IRR       Suff.  2*Pr(Suff.)     [95% Conf. Interval]

-------------+-------------------------------------------------------------

timeticabe~g |   .9751322         365      0.0262      .9526267    .9970661

    dobutrex |   1.135581       17.76      0.5162       .811387    1.511731

---------------------------------------------------------------------------

---------------------------------------------------------------------------

hprtimepoi~s |        IRR       Suff.  2*Pr(Suff.)     [95% Conf. Interval]

-------------+-------------------------------------------------------------

timeticabe~g |   .9663937         365      0.0034       .941191    .9893308

      simdax |      44752         .75      0.0142      9.535629    7.66e+07

---------------------------------------------------------------------------

---------------------------------------------------------------------------

hprtimepoi~s |        IRR       Suff.  2*Pr(Suff.)     [95% Conf. Interval]

-------------+-------------------------------------------------------------

timeticabe~g |   .9786942         365      0.0614      .9559371    1.001019

       curam |   3.469221           9      0.0160      1.251505    9.323602

---------------------------------------------------------------------------

---------------------------------------------------------------------------

hprtimepoi~s |        IRR       Suff.  2*Pr(Suff.)     [95% Conf. Interval]

-------------+-------------------------------------------------------------

timeticabe~g |   .9718895         365      0.0089      .9500632    .9930202

   sedacoron |   1.013099           6      1.0000      .3119419    2.884679

---------------------------------------------------------------------------

Methods Section (P3/L99-105): “We used exact Poisson regression to estimate the effect of the overlap-time of ticagrelor and cangrelor co-administration in minutes and potential confounding co-variables on the number of subsequent HPR episodes, expressed as incidence rate ratio (IRR) with a 95% confidence interval (95%CI). Co-variables included co-administered drugs, blood pressure levels (mmHg), heart rate (bpm), left ventricular systolic function (normal; mild, moderate, severe dysfunction) as well as pro-BNP levels (pg/ml), liver enzyme (U/l) and bilirubin levels (mg/dl) at the time of transition from cangrelor to ticagrelor.”

Results Section (P6,7/L154-156): “There was a significant relationship between the overlap-time of ticagrelor and cangrelor co-administration and the number of subsequent HPR episodes with an IRR of 1.03, 95%CI 1.01-1.05; p=0.005 (figure 4). The effect remained unchanged after adjustment for co-variables (supplement). “

Limitations Section (P8/L207-216): “The study is limited by its observational design and the small sample size of a highly selected patient cohort consisting of hypothermic cardiac arrest survivors with STEMI. We assessed platelet reactivity as a surrogate marker of thrombotic complications. The study was not powered for clinical outcomes. Compensation for potential confounding was attempted by adjustment for clinically plausible co-variables. Multivariable analyses including each relevant co-variable separately did not indicate relevant confounding. However, appropriate caution must be exercised when interpreting our results, which do not allow any conclusions to be drawn for clinical practice. Furthermore, the study only included patients who were transitioned from cangrelor to the oral P2Y12 inhibitor ticagrelor. From our results no inferences must be made as to platelet reactivity or potential drug-drug interactions after transitioning from cangrelor to clopidogrel or prasugrel.”

NB: Reviewer#3 suggested to provide a sample size calculation:

 We did not perform a formal hypothesis test based sample size calculation, but estimated our sample size on expected precision of the estimates. Assuming an average event rate of 0.5 we calculated exact Poisson 95% confidence intervals and plotted these against sample size (exposure) [1-4]. A sample size of 16 individuals turned out to be the optimum value located at the steep and flat part of the precision vs. sample size curve (please see the attached pdf to see the figure)

References:

Flierl, U.; Rontgen, P.; Zauner, F.; Tongers, J.; Berliner, D.; Bauersachs, J.; Schafer, A. Platelet inhibition with prasugrel in patients with acute myocardial infarction undergoing therapeutic hypothermia after cardiopulmonary resuscitation. Thromb Haemost 2016, 115, 960-968 Franchi, F.; Rollini, F.; Rivas, A.; Wali, M.; Briceno, M.; Agarwal, M.; Shaikh, Z.; Nawaz, A.; Silva, G.; Been, L., et al. Platelet Inhibition With Cangrelor and Crushed Ticagrelor in Patients With ST-Segment-Elevation Myocardial Infarction Undergoing Primary Percutaneous Coronary Intervention. Circulation 2019, 139, 1661-1670 Parodi G, Valenti R, Bellandi B, Migliorini A, Marcucci R, Comito V, et al. Comparison of prasugrel and ticagrelor loading doses in ST-segment elevation myocardial infarction patients: RAPID (Rapid Activity of Platelet Inhibitor Drugs) primary PCI study. Journal of the American College of Cardiology 2013; 61: 1601-1606. Holm M, Tornvall P, Henareh L, Jensen U, Golster N, Alstrom P, et al. The MOVEMENT Trial. Journal of the American Heart Association 2019; 8: e010152.

Round 2

Reviewer 3 Report

The revised manuscript has improved and the authors answered the reviewers’ concerns sufficiently.

Author Response

Thank you very much. We greatly appreciate it.